# Assessment of Patient Treatment and Rehabilitation Processes Using Electromyography Signals and Selected Industry 4.0 Solutions

**DOI:** 10.3390/ijerph20043754

**Published:** 2023-02-20

**Authors:** Ewa Stawiarska, Maciej Stawiarski

**Affiliations:** 1Faculty of Organisation and Management, Silesian University of Technology, 44-100 Gliwice, Poland; 2Faculty of Medical Biotechnology, Medical University of Lodz, 90-419 Łódz, Poland

**Keywords:** process management, treatment and rehabilitation, electromyography signal, Industry 4.0, e-health

## Abstract

Funding treatment and rehabilitation processes for patients with musculoskeletal conditions is an important part of public health insurance in European Union countries. By 2030, these processes will be planned in national health strategies (sequential process activities will be identified, care packages will be defined, service standards will be described, roles in the implementation of activities will be distinguished). Today, in many countries of the world (including the EU countries), these processes tend not to be very effective and to be expensive for both patients and insurance companies. This article aims to raise awareness of the need for process re-engineering and describes possible tools for assessing patient treatment and rehabilitation processes (using electromyographic signals—EMG and selected Industry 4.0 solutions). This article presents the research methodology prepared for the purpose of process evaluation. The use of this methodology will confirm the hypothesis that the use of EMG signals and selected Industry 4.0 solutions will improve the effectiveness and efficiency of treatment and rehabilitation processes for patients with musculoskeletal injuries.

## 1. Introduction

Patient treatment and rehabilitation is an important part of public health insurance and a key measure in achieving Sustainable Development Goal 3—“Ensure healthy lives and promote well-being for all at all ages” [1]. The treatment and rehabilitation of patients with musculoskeletal conditions as well as the promotion of good physical health and the prevention of musculoskeletal conditions account for an average of 13% of public health insurance expenditure in EU countries. Kujawa draws attention to the need to develop standards of professional practice in comprehensive treatment and rehabilitation and the need to ensure harmonisation of standards in the European Union [2]. Improving the effectiveness and efficiency of treatment and rehabilitation processes will benefit both individuals and society as a whole. Today, various medical institutions recommend the management of treatment and rehabilitation processes [3,4,5]. Initial system recommendations for the re-engineering of treatment and rehabilitation processes are identified in the Recommendations for strengthening rehabilitation in health systems [1].

Re-engineering is particularly necessary for rehabilitation processes. It is estimated that 2.4 billion people worldwide currently suffer from musculoskeletal conditions and need rehabilitation (Global Burden of Disease studies from the Institute of Health Metrics). The demand for rehabilitation is expected to increase worldwide due to changes in health status and population characteristics. People live longer but are more likely to suffer from chronic diseases and disability. Today, the need for rehabilitation is largely unmet. In some low- and middle-income countries, more than 50% of people do not receive the rehabilitation services they need. Rehabilitation services are also among those health services that have been most affected by the COVID-19 pandemic.

The treatment and rehabilitation of an orthopaedic patient aim to achieve full mobility and a return to an active lifestyle. This goal can be reached through active collaboration between the physician, diagnostician and physiotherapist. An important aspect of orthopaedic treatment and rehabilitation is to assess the progress of recovery by monitoring the patient’s body, e.g., during rehabilitation training (preferably with modern equipment that uses EMG signals) [6,7,8]. By using EMG tests and selected Industry 4.0 tools, it is possible to increase process effectiveness significantly (reduce treatment and rehabilitation time) and improve efficiency (reducing the cost of public expenditure or, if the patient does not use it, the public service, reducing his/her expenditure on private treatment).

The treatment and rehabilitation of the musculoskeletal system should be highly patient-centred, as it depends on many factors and needs individual modification of the recovery process. The existing system of treatment and rehabilitation does not allow individualisation and control of the processes taking place in multiple facilities. The patient makes their own appointments at outpatient clinics, hospitals and rehabilitation centres, always submitting medical records to healthcare professionals. Medical staff lose time in reviewing medical records. Sometimes the patient changes medical institution/physician and, consequently, tests are duplicated, and sometimes are incomplete. Recovery is difficult and time-consuming for the patient and requires patience and funding. Funding (excluding funding for treatments in public centres) for illness and rehabilitation is provided by the insurance companies post factum (i.e., after the documentation of illness has been presented, which sometimes takes years). This period is filled with anxiety about one’s own health, full of pain, lack of money and know-how, which are not conducive to recovery [9].

## 2. Materials and Methods

### 2.1. Materials 1. Applicable Treatment and Rehabilitation Processes for a Patient with a Musculoskeletal Injury

The patient’s treatment and rehabilitation processes take place in various medical facilities (inpatient hospitals, outpatient departments, sanatoriums, private clinics and the patient’s home). The staff involved in the process consist of a variety of healthcare professionals, including but not limited to physiotherapists, occupational therapists, orthopaedists and orthotists, clinical psychologists, physical and rehabilitation physicians and rehabilitation nurses. Each medical facility and even an individual healthcare professional have an effect on the effectiveness and efficiency of processes. In the age of the development of medical engineering and Industry 4.0, however, these indicators can be improved further.

Two processes can be distinguished: treatment and rehabilitation. The treatment process starts with the injury and its diagnosis, or with a diagnostic evaluation that identifies degenerative changes. Sometimes a surgical procedure is necessary, followed immediately by rehabilitation to provide an easier postoperative period.

If the injury or degeneration does not require surgical treatment, conservative rehabilitation is used. A consultation with a physiotherapist starts the rehabilitation process. An effective rehabilitation process consists of pain relief followed by restoration of mobility and muscle strengthening. The later stage of rehabilitation involves injury prevention through specialised functional exercises and proprioception.

In the above description of the processes of treatment and rehabilitation of the musculoskeletal system, further measures/tasks faced by healthcare can be distinguished. However, in the turbulent reality experienced by healthcare and patients after the COVID-19 pandemic, it is difficult to find standard measures, and there is a lack of follow-up procedures for the aforementioned processes. Meanwhile, insurance companies seem to look for savings on patient treatment and rehabilitation processes.

### 2.2. Materials 2. EMG Signal in the Treatment and Rehabilitation of Patients with a Musculoskeletal Condition

Biomedical signals make it possible to monitor the status of the patient’s body. For example, an electrocardiogram (ECG) verifies the state of the heart. Other similar examinations include [10]:Electroencephalography (EEG);Electromyography (EMG);Electrooculography (EOG);Electrogastrography (EGG).Biological signals can be divided according to their source into [11]:Bioelectrical (ECG, EEG, EMG);Bioimpedance (tissue impedance measurement);Bioacoustic (voice, heart sounds);Biomagnetic (measurement of the magnetic field generated by internal organs, e.g., the brain, heart, lungs);Biomechanical (musculoskeletal diagnosis, mechanical heart rate);Bio-optical (e.g., oximetry), other (e.g., spirometry).

The biofeedback concept uses various signals to reflect the physiological activity of the human body, so that the user (physician, therapist, patient) can obtain information (which cannot be manipulated) about the state of selected organs. Following behavioural stimulation supplemented by feedback, devices suggest a therapeutic concept. Some of the devices have a therapeutic effect. Long-term therapy with feedback allows the best therapeutic effect to be achieved.

In the medical biofeedback concept, an EMG signal is used to detect any condition associated with damage to the peripheral nervous system or muscular system. Motor disabilities can have two main causes: a damaged nervous system—an inability to transmit the nerve impulse that stimulates the muscles, or a damaged muscle that moves the limb. Unfortunately, the effect is the same in both cases—motor disability. EMG testing can detect the cause of the lack of motor skills of the musculoskeletal system [12]. EMG signals allow an appropriate diagnosis to be made and appropriate therapy to be implemented. The most common type of therapy for motor disabilities is rehabilitation, which can be supported by EMG signals. The EMG signalling device supports not only patients but also athletes. The analysis of the EMG signal while training allows abnormalities in the exercise performed to be detected. The EMG signal can also be used to convert sign language into human speech.

State-of-the-art EMG information and communication technology are based on biofeedback. This technology can be used for the enhancement of normal movement patterns after injury and the better monitoring of home-based autotherapy [13]. Biofeedback systems can provide important information regarding exercise technique and quality, allowing real-time movement corrections [14]. The use of digital biofeedback yields better results than conventional physiotherapy [15,16].

A portable system for muscle rehabilitation is one of the biofeedback devices that use an EMG signal. Invasive and non-invasive methods are used to record signals [17]. The main difference is in the way the electrodes are arranged and attached, by means of which the signal is transmitted to the amplifier. Invasive methods involve attaching electrodes directly to the medium from which the signal is taken. In the case of EMGs, it is the insertion of electrodes into muscles and the recording of nerve impulses sent to the muscle. The second method is non-invasive. Electrodes are attached to the muscle area on the skin. This method is less accurate, as the signal recorded is the sum of the signals of each nerve fibre from around the electrodes. This problem can be solved by appropriate signal filtering.

The portable system for muscle rehabilitation is easy to carry and allows the physician or therapist to monitor the patient’s muscles in a remote manner. By using this device, the patient can perform rehabilitation exercises in more places (even at home), their posture is more relaxed and the interaction of the device means that the long-term treatment process is no longer burdened by errors [18]. The device includes a surface myoelectric sensor, signal transmitter, rhythm generator, comparator and renderer. The surface myoelectric sensor is configured to collect the user’s electromyographic signals. The signal transmitter is connected to the surface electromyographic sensor and is configured to receive and transmit the myoelectric signal. The rhythm generator is configured to generate a specific rhythm. The comparator is connected to the signal transmitter and the rhythm generator and is configured to receive the myoelectric signal (i.e., the rhythm emitted by the user’s muscle). The rhythm generated and rhythm emitted are compared to determine differences. The renderer is connected to the rhythm generator and to the comparator, and is configured to receive and present a specific rhythm to the user. When the EMG signal shows positive feedback, it means that the rhythm is consistent. Negative feedback is demonstrated when the myoelectric signal does not match a specific rhythm. As an option, the portable EMG signal-based system for muscle rehabilitation includes [19,20,21]:A signal processor connected between the surface myoelectric sensor and the signal transmitter;An electrode sensor, a signal amplification circuit connected to the electrode sensor and a signal connected to the signal amplification circuit. The circuit and the signal smoothing circuit are connected to the signal full-wave rectifying circuit;The electrode sensor includes a reference electrode, a muscle intermediate electrode and a muscle end electrode;The signal processor contains an A/D converter and a digital signal processor, and the A/D converter is connected to the connector of the digital signal processor;A remote monitor coupled to the renderer and configured to receive feedback from the renderer.

All the optional elements listed and those not mentioned above support the monitoring and recording of signals from the affected muscle or muscle group [22]. The portable EMG signal-based system for muscle rehabilitation allows the user to continuously strengthen and exercise a target muscle or muscle group using the cardiac rhythm device, comparator and renderer. The high degree of interaction with the device allows the user to control the monitored muscles and suppress inappropriate muscle contractions. The compiled feedback results show the user’s muscle recovery status. For identified conditions that do not resolve, the device suggests other treatment and exercise sets.

The EMG signal is a bioelectrical signal that is analogue in its original form. It is digitised so that it can be used and analysed by computers. In the portable EMG signal-based treatment system for muscle rehabilitation, the signal transmitter is a wireless transmitter that works with a computer via Bluetooth, infrared transmitter or Wi-Fi. Energy saving is an important feature of the transmitter. Extremely low power consumption during operation and standby can make a button cell last for several years. Its main advantages include very low peak, medium and standby power consumption; low price; increased wireless coverage; full backward compatibility and low latency. Signals can be transmitted to servers or other computers using special software [23].

The portable system for muscle rehabilitation can send information from the human body via a computer, and this information can be recorded and stored using new Industry 4.0 solutions, thereby improving the patient’s treatment and rehabilitation processes.

### 2.3. Materials 3. Industry 4.0 Tools in the New Treatment and Rehabilitation Processes for Patients with a Musculoskeletal Injury

Blockchain technology is constantly being tested for use in data storage or data protection in a wide variety of industries and sectors. In addition to industries such as charity and the supply chain, healthcare is one of the most discussed use cases for blockchain networks. There are many aspects of blockchain that make the blockchain technology suitable for the healthcare industry. As most blockchain networks are designed as distributed systems that record and protect files using, among other things, cryptography, it is extremely difficult for anyone to launch an effective attack against them or alter the data stored in them without obtaining the consent of all other network participants. Immutability is the very feature of blockchains that offers the possibility of creating secure medical databases. Furthermore, the peer-to-peer (P2P) architecture used in blockchain networks allows all copies of patient’s medical records to be synchronised with each other when they are updated, even if they are stored on different computers and in different locations. In fact, each network node has a copy of the entire blockchain and regularly communicates with other nodes to ensure that this copy is up to date and its data are authentic. Decentralisation and distributed data distribution are other important and useful features in the healthcare industry. Blockchain networks, despite being distributed, are not necessarily decentralised (in terms of management, e.g., an insurer may be authorised to manage data i.e., effectively record and track the medical data of thousands of patients). In terms of healthcare, blockchain networks most often function as private networks (requiring appropriate rights to be obtained). Unlike traditional databases, which are usually based on a single centralised server, the use of a distributed system allows data to be exchanged while achieving a higher level of security and reducing administrative costs. The decentralised nature of blockchain networks makes stored data less susceptible to technical failures or external attacks. The security provided by blockchain networks can be particularly useful in hospitals that face hacker attacks or malware such as ransomware.

Another advantage of relying on a blockchain for medical records is its ability to improve interoperability between clinics, hospitals and other healthcare providers. Technological differences in data storage systems often make it difficult for organisations to share collected documentation with each other. Blockchains can solve this problem by allowing authorised parties/persons to access a unified database containing all the information and files on patients, or even the history of medicines prescribed to them. This gives service providers the opportunity to undertake mutual collaboration and exchange the data they need. In addition to simplifying the process of sharing medical records, image files and other patient records, blockchain networks can also provide patients themselves with a higher level of accessibility to and transparency of the information collected about them. By implementing a blockchain, unauthorised changes to patient records and the risk of typical human errors, such as spelling mistakes, are eliminated.

The blockchain technology can also be used against insurance fraud, e.g., claiming compensation for situations that never happened. From the patient’s perspective, the application of blockchain technology will save them from having to complete documents to prove health events to the insurer (this will also reduce expenses for expert reports and legal services).

Another potential use of blockchain in healthcare is for improving the quality and effectiveness of tests, treatment and rehabilitation. Medical data stored in blockchains can be encrypted and then used for:Identifying patients who could benefit from the potential good effects of the drugs being tested;Qualifying patients for medical treatment and rehabilitation based on the urgency of needs;Monitoring the progress of treatment and rehabilitation.

In recent years, some blockchain-based medical data sharing systems have been used by the academic community. Encrypted patient data are used for medical science and education. However, information privacy and accessibility are still problematic [24]. A public blockchain (without patient data) that is available to all would support patients seeking information about similar medical cases.

A private blockchain (i.e., the one with visible patient data) also needs some thought. The designated organisation to create blockchain alliance needs to rethink the data authentication plan. This is because the following questions arise: Can each participant have a backup of all patient data? Can any participant be involved in data recording and storage? Should the patient authorise access to data in the cloud each time? The idea is that in a distributed P2P network without a central control node (by means of an algorithm), a self-organising network is established. Such a blockchain solution has many advantages, such as decentralisation, fast distribution, limited data manipulation and traceability [25]. Work on the real-time monitoring of the treatment process and the possibility of making comparisons of the treatment and rehabilitation processes of different patients is also underway. The majority of work related to data validation in a blockchain is still in the experimental phase [26,27]. There has been more progress on the development of a blockchain for treatment process monitoring. This is because it is more important to limit medical resources and reduce treatment costs [28]. Given the problem, Chen et al. [29] as well as Wang [30] and Meng et al. [31] prepared models for blockchain-based medical data sharing systems.

### 2.4. Methods. Research Methodology for Evaluating the Treatment and Rehabilitation Processes for a Patient with a Musculoskeletal Injury

The main objective set for the research model was to evaluate the treatment and rehabilitation processes for patients with a musculoskeletal injury.

The subject of the study was the processes of treatment and rehabilitation of the musculoskeletal system. The next steps in the treatment and rehabilitation processes are listed in Table 1. The course of the processes was verified after the analysis of the medical records of patient examinations. The detailed processes were evaluated before and after their re-engineering (i.e., after the use of state-of-art EMG-based biofeedback devices and Industry 4.0 tools).

Research subjects: The research subjects included two groups of purposively selected patients. The main selection criteria included sex—women were studied; age—female patients aged 50 (plus or minus 5 years) were studied; type of injury—female patients with a historical ankle joint injury in the form of a trimalleolar fracture (lateral ankle, medial ankle and posterior edge of tibia) were studied. In the first group of patients (15 women), there were those who were not supported with an EMG-based biofeedback device during treatment and rehabilitation. The second group of patients (15 women) included those who were assisted with an EMG-based biofeedback device during treatment and rehabilitation. It was also important that the patients consciously monitor the course of their condition, i.e., they remember the medical facilities providing assistance, measure the recovery time of the joint and muscles of the limb and document the expenses they incurred during treatment (reimbursed and not reimbursed by the insurer).

Research methods: Two research methods were used:The analysis of the medical records of the patients to classify them for the study and to clarify the possible course of treatment and rehabilitation processes for the patients with a selected disease entity;Diagnostic survey, technique: questionnaire, research tool: survey questionnaire completed face-to-face by an interviewer.

During the creation of a model of the treatment and rehabilitation processes (Table 1), strategic activities (diagnostic tests) and possible surgical procedures as well as supporting activities for treatment and rehabilitation (hospitalisation, outpatient treatment, rehabilitation, e-rehabilitation, interdisciplinary consultations) were identified. The survey questionnaire outlined to the participants the treatment and rehabilitation processes for their injury (the participants were able to detail this process). The detailed process was also supplemented by reference treatment and rehabilitation processes developed by the Polish School of Rehabilitation [31]. Based on the recommendations of Marian Weiss and Aleksander Hulek, who made an important contribution to the development of comprehensive rehabilitation, Weiss postulated that the rehabilitation process should involve the whole team, consisting of physicians, psychologists, physiotherapists and counsellors (depending on the type of injury). Hulek, on the other hand, contributed to the scientific basis of comprehensive rehabilitation and the adoption of its model by the WHO in the 1970s. In the preparation of the process evaluation measures, the following studies were used:Miller, Pniewski and Polakowski (2000), who suggested a subjective three-grade assessment of the value added in each activity; Wee and Wu (2009) [32], who mapped value streams, Wynstra et al. (2008) [33], who assessed the value added by process activities using a 5-point Likert scale.Eisenhardt and Martin (2000), who suggested the measurement of process times [34]. Behrouzi et al. (2011), who considered that time is a standard indicator of process performance [35].Wilden et al. (2016), who suggested measuring risks [36].

The survey questionnaire presented a detailed treatment and rehabilitation process with columns to allow the patient to assess the effectiveness and efficiency of the process (Table 2).

Data were collected using a form (from two groups of patients). The quantitative data were summarised and averaged. The processes whose selected steps were assisted by the EMG-based feedback device were carried out in the second group of participants. Following the assessment of the processes carried out in the first and second groups, an additional simulation evaluation was carried out (assuming the application of blockchain and Big Data technologies in these processes). The simulation was carried out using the form shown in Table 2 and Table 3.

## 3. Results

A comprehensive assessment of treatment and rehabilitation processes (following implementation of an EMG-based biofeedback device) can be an important achievement for the re-engineering of these processes (taking place in the health systems of EU countries). This study aims to assess the course of these processes before and after the implementation of the innovation. According to Gunasekaran et al. (2005), the main dimension of innovation performance is the number and quality of solutions implemented in the final process; however, in the case of treatment and rehabilitation processes, their effectiveness and efficiency are more important [37].

Figure 1, Figure 2, Figure 3 and Figure 4 show the assessments of the processes in question in terms of:Efficiency (measures: the costs incurred by the insurer and the patient in the treatment and rehabilitation processes, and the number of risks affecting the efficiency of the processes);Effectiveness (measures: creation of values for the patient’s physical and mental health, and number of risks that also affect the patient’s health).

The respondents gave similar ratings for the DL1 (exploratory diagnosis and identification of specialists for consultation) stage. All assessment criteria, i.e., cost, value for mental health, time and identified risks, received the same rating. This is because the respondents were injured (meaning that they were not subjected to EMG signals at the DL1 stage and did not use Industry 4.0 solutions to prepare for treatment). Similar studies conducted among doctors would probably show differences in the assessment of the DL1 stage for the criteria of cost, value for the patient’s mental health, time and identified risks. It is planned to conduct similar studies among doctors and physiotherapists in the future.

## 4. Discussion

The analysis of the collected data, shown in Figure 1, supports the hypothesis that the use of EMG signals and selected Industry 4.0 solutions will improve the efficiency of treatment and rehabilitation processes for patients with musculoskeletal injuries. The highest process costs were incurred by carrying out processes in which no innovations were implemented. The averaged data showed the total costs of the treatment and rehabilitation processes (for the selected condition) to be PLN 6000, i.e., approximately EUR 1500. The lowest cost can be achieved when all the proposed innovations are implemented, and their cost is EUR 800.

The analysis of the collected data, shown in Figure 2, supports the hypothesis that the use of EMG signals and selected Industry 4.0 solutions will improve the effectiveness of treatment and rehabilitation processes for patients with musculoskeletal injuries. The highest process scores were achieved by simulating the implementation of all the innovations in question. It should be noted that the patients who used the EMG-based biofeedback consciously implemented the rehabilitation process (they were able to detail it, e.g., by listing three steps and different types of exercise). The first group of subjects (no innovations in the treatment and rehabilitation processes were implemented in this group) were not aware of the course of rehabilitation process, i.e., they performed it chaotically and for a long time without results.

The analysis of the collected data, shown in Figure 3, supports the hypothesis that the use of EMG signals and selected Industry 4.0 solutions will improve the effectiveness of the treatment and rehabilitation processes of patients with musculoskeletal injuries, also in the area of mental health. The highest process scores were achieved by simulating the implementation of all the innovations in question. It should be noted that the patients who used the EMG-based biofeedback consciously implemented exercises and monitored their progress towards physical recovery. That awareness was reflected in their mental health. Many of the participants pointed out the lack of examples and data on the progress of treatment and rehabilitation in the selected disease entity. Knowledge of the condition, stored in large data sets, would negate the fear and anxiety that accompany slow recovery.

The analysis of the collected data, shown in Figure 4, supports the hypothesis that the use of EMG signals and selected Industry 4.0 solutions will improve the effectiveness and efficiency of treatment and rehabilitation processes for patients with musculoskeletal injuries. The lowest number of risks in the process was recorded by simulating the implementation of all the innovations in question. The most important risks that arise in the treatment and rehabilitation processes are reviewed below. The most frequently cited risks occurring in a process to which no innovation was implemented (EMG-based biofeedback devices, blockchain and Big Data) are:Diagnostic problems in patients with co-morbidities or diseases acquired during treatment of the underlying injury (e.g., neurological diseases, cardiovascular diseases).Doubts/discussions regarding the diagnosis of the interdisciplinary case conference.Doubts/discussions regarding the use of preoperative rehabilitation.Delegation of tasks/problems with designating medical or rehabilitation staff.Patient doubts about the treatment given/lack of full internal patient involvement.Long, uncoordinated postoperative recovery period/overdue rehabilitation appointments.Necessity of repeating diagnostic evaluation and return to the treatment process.Error in choosing a rehabilitation facility that does not specialise in treating the condition.Lack of progress in recovery at the health facility chosen by patient/poor atmosphere/lack of empathy.Strain during rehabilitation.Unexpected/unwanted emotional problems in social, occupational and psychological rehabilitation.Willingness to change the rehabilitation facility during the project.Numerous changes in the composition of consulting physicians who monitor the results of treatment and rehabilitation.In rehabilitation, scheduled hours and appointments reimbursed by the insurer were overrun.Unclear, restrictive attitude towards the patient/lack of standardisation of patient management.Incomplete access to critical information regarding payments by the insurer.Relapse of the formally cured disease.

The above-mentioned problems represent the largest group of risks mentioned in treatment and rehabilitation processes without innovation. They are also usually the most severe and have the greatest impact on the effectiveness and efficiency of treatment. The assessment of the process by the number of risks occurred is not authoritative. Therefore, an individual risk assessment is recommended. Generally, the highest incidence of specific problems in processes without innovation may be indicative of weaknesses in the management of the current health system.

However, studies reveal that the implementation of biofeedback into medical processes also carries risks, e.g.:Problems with e-rehabilitation devices, their lack/lack of funding, problems with the communication interface.Lack of staff to remotely monitor the convalescent’s progress.

Currently, the development of a blockchain and Big Data for the assessment and support of rehabilitation processes also faces several problems to be solved. These include [38]:Transparency/standardisation of data on rehabilitation services, assessment of rehabilitation and post-rehabilitation effectiveness, assessment of rehabilitation facilities, assessments of individual rehabilitation services.The effortfulness associated with the need to describe the cases of rehabilitation patients, the time delays involved, the high error rate, the high labour costs, the lack of standardisation of the documents prepared.Privacy, which is more difficult to provide in a rehabilitation facility. Original documents are available to numerous staff members.

## 5. Conclusions

Treatment and rehabilitation processes can be defined as a comprehensive action aimed at restoring as much mobility and ability to function independently in society as possible to a person who has lost this ability as a result of an illness or injury. A coordinated effort between physicians, rehabilitation specialists, psychologists, physiotherapists and vocational counsellors is needed for the comprehensive rehabilitation process to be implemented properly. The foundations of comprehensive rehabilitation were laid as early as the 1950s by Wiktor Dega, an orthopaedic surgeon. This model was considered by the World Health Organisation to be worthy of imitation and promotion. The Polish School of Rehabilitation included four basic assumptions as part of the characteristics of the rehabilitation process:Universality—assuming that everyone has the right to rehabilitation, regardless of diagnosis, age or prognosis;Earliness—starting rehabilitation at the earliest possible stage, from the first day in hospital;Comprehensiveness—comprehensive coordinated actions carried out by a team of specialists, targeting all levels of the person’s life and responding to his/her individual needs;Continuity—the systematic provision of rehabilitation interventions and continuing them for as long as a given person needs them.

Comprehensive rehabilitation consists of four different types of rehabilitation: medical, psychological, social and occupational. The model of comprehensive rehabilitation developed by the Polish School of Rehabilitation took hold in many European countries and would still be used today if it had not been for the lack of procedures and funding. Comprehensive rehabilitation must undoubtedly be defined in the form of a process (i.e., a sequence of activities). It is not, only because the standardisation of the activities of this process (universality, comprehensiveness, earliness, continuity) would exceed the costs of current insurance systems. The implementation of this process into the social security system is an investment in people. This investment will be several times more than compensated. Fortunately, the WHO has taken care of future changes and re-engineering of treatment and rehabilitation processes. The guidelines by the WHO include [1]:Prepare for situation assessment;Collect data and information;Conduct assessment in the country;Write, revise and finalise report, disseminate and communicate findings;Prepare for strategic planning;Consult, revise, finalise and complete costing of plan;Identify priorities and produce first draft of plan;Endorse and disseminate the strategic plan;Develop monitoring framework with indicators, baselines and targets;Establish evaluation and review processes;Establish a recurring implementation “plan, do, evaluate” cycle;Increase capacity of rehabilitation leadership and governance (Figure 1).

Treatment and rehabilitation processes can be re-engineered by implementing innovative inventions of biomedical engineering and Industry 4.0. The advantage of re-engineering treatment and rehabilitation processes by implementing innovations is the improvement in effectiveness (health status improvement) and efficiency (cost reduction) of such processes. The innovations described in this paper enable process improvements by:Conducting remote rehabilitation [11,39,40];Focusing on patients by keeping detailed records of their health status;Digitising treatment and rehabilitation information.

Rehabilitation information is an important reference point for monitoring and updating the patient’s health status. Big Data will enable the continuous advancement of data mining technology, can effectively extract diagnostic information regarding cured patients and make comparisons of the data with other similar data, which is helpful for physicians to develop a treatment and rehabilitation process [41].

The innovations described in this paper make it possible to improve process efficiency by:Online medical and rehabilitation appointment scheduling, etc.;Reducing data storage costs;Storing and sharing data that increasingly contribute to the efficiency of treatment and rehabilitation processes [28];Reducing costs for individual patients, which is particularly important for patients with reduced mobility or who live far from health centres [12].

## Figures and Tables

**Figure 1 ijerph-20-03754-f001:**
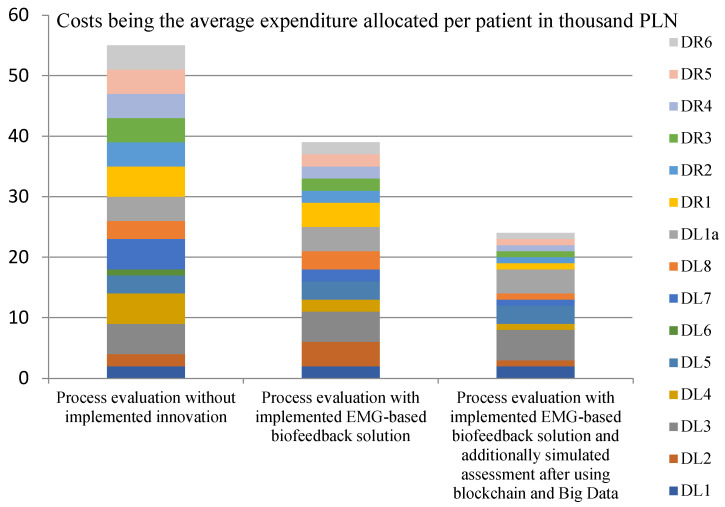
Assessments of the process steps by cost. Source: Authors’ own elaboration based on primary studies.

**Figure 2 ijerph-20-03754-f002:**
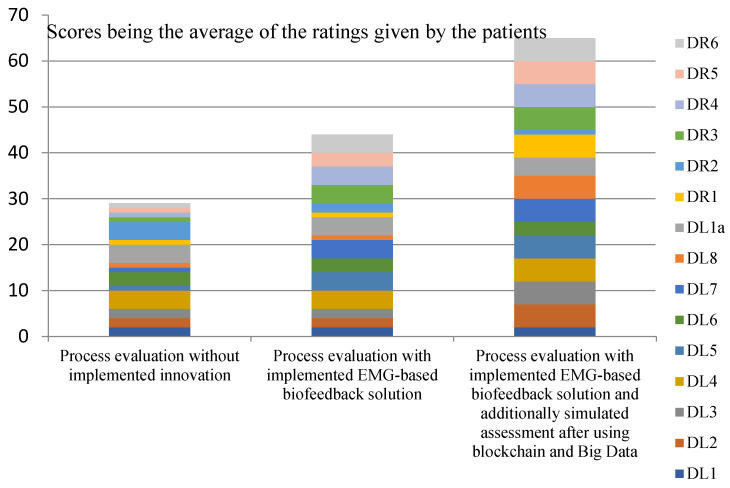
Assessments of the process steps in terms of the creation of values for the patient’s physical health. Source: Authors’ own elaboration based on primary studies.

**Figure 3 ijerph-20-03754-f003:**
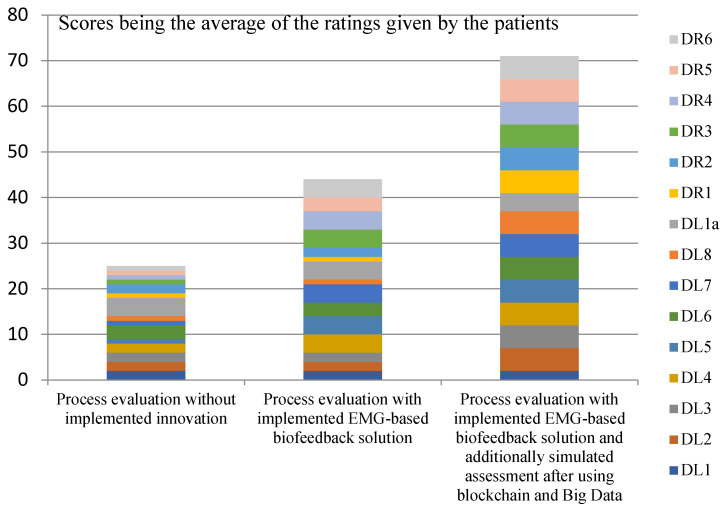
Assessments of the process steps in terms of the creation of values for the patient’s mental health. Source: Authors’ own elaboration based on primary studies.

**Figure 4 ijerph-20-03754-f004:**
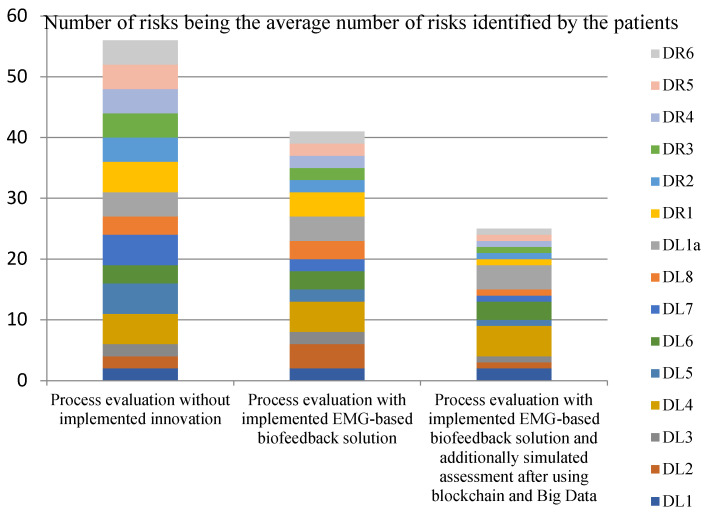
Assessments of the process steps by the number of risks identified. Source: Authors’ own elaboration based on primary studies.

**Table 1 ijerph-20-03754-t001:** Steps in the treatment and rehabilitation processes of a selected injury (trimalleolar fracture of ankle joint).

Steps of Treatment and Rehabilitation Processes	Description of the Activity Performed during the Process Step
DL1:	Exploratory diagnosis and identification of specialists for consultation
DL2:	Case conference with a rehabilitation specialist
DL3:	Observation and preparation of the patient for surgery, including communicating treatment and management procedures and performance standards (core staff and medical psychologist)
DL4:	Hospitalisation—osteosynthesis
DL5:	Monitoring of treatment progress (sometimes using e-rehabilitation); return to DL1 if necessary
DL6:	Syndesmotic screw removal surgery
DL7:	Monitoring of treatment progress (sometimes using e-rehabilitation); return to DL1 or termination of the treatment process if necessary
DL8:	Completion of treatment and payment of benefits based on documents collected during the treatment process
DR1:	Scheduling appointments and recommendation of medical rehabilitation providers
DR2:	Monitoring of the medical rehabilitation progress (post-rehabilitation diagnostic evaluation including the use of e-rehabilitation devices) (return to DR1 if necessary)
DR3:	If necessary, scheduling appointments and recommendation of providers of social, occupational and psychological rehabilitation services
DR4:	Monitoring the progress of social and occupational rehabilitation
DL1a	Resumption of treatment/Hospitalisation—procedure to remove internal fixations
DR5	Monitoring of the medical rehabilitation progress (post-rehabilitation diagnostic evaluation including use of e-rehabilitation devices) (return to DL1a if necessary)
DR6	Completion of rehabilitation, health impairment price and compensation of benefits paid based on documents collected during the rehabilitation process.

Source: Authors’ own elaboration.

**Table 2 ijerph-20-03754-t002:** A detailed form for assessing the steps that make up the processes of treatment and rehabilitation of patients following a specific injury, i.e., trimalleolar fracture of ankle joint.

Next Process Step	Duration of Step	Costs Incurred by the Insurer or the Patient (Own Costs Could Be Reimbursed by the Insurer)	Added Value Created		Number of Identified Risks	Activity Support Tools
			Value for Patient Physical Health Points 1–5 1—Lowest Score 5—Highest Score	Value for Patient Mental Health Points 1–5 1—Lowest Score 5—Highest Score		
DL1:		-	-	-		
DL2:		-	-	-		
DL3						
DL4						
DL5						
DL6						
DL7						
DL8						
DL1a						
DR1						
DR2						
DR3						
DR4						
DR5						
DR6						
	Total	Sum of money	Total points	Total points	Sum of risks	

Source: Authors’ own elaboration based on: Miller JA, Pniewski K, Polakowski M, (2000), Zarządzanie kosztami działań, Wig-Press Artur Andersem, Warsaw. The rows (DL1-DL1a) in white in the table indicate the stages of the process of treating the patient. The gray rows (DL1-DL1a) in the table indicate the stages of the patient’s rehabilitation process.

**Table 3 ijerph-20-03754-t003:** Form for the simulation evaluation of the process. The assessment was made after possible implementation of blockchain and Big Data technologies.

Process Step	Process Stakeholders’ Activities after Blockchain Application			Products of Various Steps of the Process after Blockchain and Big Data Implementation
	Authentication centre	Health facilities as: - Pre-processing node - Sort node - Confirmation node	Patient	
**DL1**	Archiving of documents	Preparation of documents	Acceptance of document availability Receipt of information	Determining the extent of treatment needs
**DL2**	as above	as above	as above	Determining the extent of treatment needs and preoperative rehabilitation needs/inclusion of preoperative rehabilitation
**DL3**	as above	as above	as above	Information provided by the patient on the hardship they have to endure to recover
**DL4**	as above	as above	as above	Information to prepare the physician performing the procedure. Carrying out necessary treatment procedures
**DL5**	as above	as above	as above	Coordinating follow-up visits based on the results of the patient’s remote devices that monitor patient health status. Setting necessary follow-up appointments and subsequent diagnostic measures
**DL6**	as above	as above	as above	Information to prepare the physician performing the procedure. Carrying out necessary treatment procedures
**DL7**	as above	as above	as above	Setting therapy appointments and rehabilitation therapy plan, online support of the patient in performing exercises
**DL8**	as above	as above	as above	Comprehensive information for the insurer on the patient’s health status. Payment of benefit without patient involvement
**DL1a**	as above	as above	as above	Information to prepare the physician performing the procedure. Carrying out necessary treatment procedures
**DR1**	as above	as above	as above	Suggesting alternative subproviders of medical rehabilitation services
**DR2**	as above	as above	as above	Ongoing provision of information on health status and recovery progress, involvement of the patient in the self-rehabilitation process
**DR3**	as above	as above	as above	Suggesting alternative subproviders of social, occupational and psychological rehabilitation services
**DR4**	as above	as above	as above	Ongoing provision of information on health status and recovery progress, involvement of the patient in the self-rehabilitation process
**DR5**	as above	as above	as above	Setting therapy appointments and rehabilitation therapy plan, online support of the patient in performing exercises
**DR6**	as above	as above	as above	Comprehensive information for the insurer on the patient’s health status. Payment of benefit without patient involvement

Source: Authors’ own elaboration based on: Wynstra F, Van Echtelt FEA, Van Weele AJ, Duysters G, (2008), Managing Supplier Involvement in New Product Development: A Multiple-Case Study, Journal of Product Innovation Management, Vol. 25, Issue 2, pp. 1–43. The rows (DL1-DL1a) in white in the table indicate the stages of the process of treating the patient. The gray rows (DL1-DL1a) in the table indicate the stages of the patient’s rehabilitation process.

## Data Availability

Patient data collected are not publicly available.

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
