# Peer review of "Assessment of Patient Treatment and Rehabilitation Processes Using Electromyography Signals and Selected Industry 4.0 Solutions"

_ijerph, 2023, doi:10.3390/ijerph20043754_

Round 1
Reviewer 1 Report
Dear. Authors,
Overall, this research paper is very fresh and exciting as it deals with the convergence of the healthcare business and Industry 4.0 solutions. Since this research paper deals with an unfamiliar topic, I got the impression that the explanation of related concepts was a bit excessive, but it was not a big problem.
There is a spelling error (SR1) in the descriptions of rows DR2 and DR5 in Table 1. Please check and correct.
Thank you.
Author Response
Reply to reviewer
First of all, thank you very much for your positive review and good opinion.
We corrected the indicated description in the table and made a slight linguistic correction (linguistic corrections were made by a native English speaker).
Based on the feedback from both reviewers, we also made the following changes:
- references to the text in the introduction have been supplemented (they are now quotes 6,7,8,9),
- due to the introduction of references 6-9, the order of the following articles cited has also changed,
- missing markings in tables and figures have been corrected,
- it was justified why the DL1 process stage is similarly assessed by the respondents. The justification in the text reads as follows:
“The respondents gave similar ratings for the DL1 (exploratory diagnosis and identi-fication of specialists for consultation) stage. All assessment criteria, i.e. cost, value for mental health, time and identified risks, received the same rating. This is because the respondents were injured (meaning that they were not subjected to EMG signals at the DL1 stage and did not use Industry 4.0 solutions to prepare for treatment). Similar studies conducted among doctors would probably show differences in the assessment of the DL1 stage for the criteria of cost, value for the patient’s mental health, time, and identified risks. It is planned to conduct similar studies among doctors and physiotherapists in the future.”
Thank you very much once again.
Authors

Reviewer 2 Report
This work investigates that the use of EMG signals and selected Industry 4.0 solutions will improve the effectiveness and efficiency of treatment and rehabilitation processes for patients with musculoskeletal injuries. The manuscript is well written and data presented is reasonable, so I think it can be published after minor revisions as below:
1. The last three paragraphs of Introduction lack references. Please add some related references. For instance, in lines 52-55, the references related to monitoring the patient's body can be added, such as “Fang C, et al. EMG-centered multisensory based technologies for pattern recognition in rehabilitation: state of the art and challenges[J]. Biosensors, 2020, 10(8): 85.”, “Hong Y, et al. Highly anisotropic and flexible piezoceramic kirigami for preventing joint disorders[J]. Science Advances, 2021, 7(11): eabf0795.” And “Kim J, et al. Wearable biosensors for healthcare monitoring[J]. Nature biotechnology, 2019, 37(4): 389-406.”
2. Please check the “DL1a” in Figure 1 and Figure 4. And the “DL1” is missing in Figure 1 and the “DL1” and “DL2” are missing in Figure 4.
3. It’s better for the authors to explain that why there is nearly no change for the DL1 group in the three process evaluations.
Author Response

(The authors gave the same response as above.)
